

# The macroevolution of size and complexity in insect male genitalia

Andrey Rudoy and Ignacio Ribera

Institute of Evolutionary Biology (CSIC-Universitat Pompeu Fabra), Barcelona, Spain

## ABSTRACT

The evolution of insect male genitalia has received much attention, but there is still a lack of data on the macroevolutionary origin of its extraordinary variation. We used a calibrated molecular phylogeny of 71 of the 150 known species of the beetle genus *Limnebius* to study the evolution of the size and complexity of the male genitalia in its two subgenera, *Bilimneus*, with small species with simple genitalia, and *Limnebius* s.str., with a much larger variation in size and complexity. We reconstructed ancestral values of complexity (perimeter and fractal dimension of the aedeagus) and genital and body size with Bayesian methods. Complexity evolved more in agreement with a Brownian model, although with evidence of weak directional selection to a decrease or increase in complexity in the two subgenera respectively, as measured with an excess of branches with negative or positive change. On the contrary, aedeagus size, the variable with the highest rates of evolution, had a lower phylogenetic signal, without significant differences between the two subgenera in the average change of the individual branches of the tree. Aedeagus size also had a lower correlation with time and no evidence of directional selection. Rather than to directional selection, it thus seems that the higher diversity of the male genitalia in *Limnebius* s.str. is mostly due to the larger variance of the phenotypic change in the individual branches of the tree for all measured variables.

## INTRODUCTION

Insect genitalia have been the focus of much attention since the middle of the 19th century, when their taxonomic value to diagnose species with otherwise very similar external morphologies was recognised. It soon became apparent that this variability should play a prominent role in reproductive isolation and speciation (e.g., *Dufour, 1848*), generating much debate on the causes of its origin and evolution (*Eberhard, 1985*; *Eberhard et al., 1998*; *Simmons, 2014*). Sexual selection is now widely acknowledged as being the major force driving the evolution of animal genitalia (*Hosken & Stockley, 2004*; *Simmons, 2014*), but there is still little consensus on what are the dominant factors (such as male competition, male–female sexual conflict or pure female choice) or the main macroevolutionary trends generating their diversity (*Simmons, 2014*).

One of the few widely accepted general trends is the negative allometric scaling of most genital traits (*Eberhard et al., 1998*; *Hosken, Minder & Ward, 2005*; *Eberhard, 2009*), which suggests that genital morphology is often subject to stabilising selection. However, genitalia are also considered to be among the fastest evolving traits in arthropods, clearly differing in

Corresponding author
Ignacio Ribera,
ignacio.ribera@ibe.upf-csic.es

**Table 1** Average values of the measured variables in the outgroup (*Laeliaena*), the reconstructed ancestor of *Limnebius*, and the reconstructed ancestor and the extant species (maximum and minimum values) of the two subgenera. *per*, perimeter; *fd*, fractal dimension; *lm*, male body length; *lf*, female body length; *lg*, length of the male genitalia.

|    | *Laeliaena* | *Limnebius* | *Bilimneus* | | | *Limnebius* s.str. | | |
|----|-------------|-------------|-------------|-----|-----|--------------------|-----|-----|
|    |             | Ancestor    | Min         | Max | Ancestor | min           | Max | Ancestor |
| *per* | 3.00     | 3.74        | 2.18        | 2.73 | 2.69 | 2.38             | 10.07 | 4.89 |
| *fd*  | 1.28     | 1.24        | 1.16        | 1.20 | 1.19 | 1.18             | 1.42  | 1.28 |
| *lm*  | 1.42     | 1.25        | 0.82        | 1.07 | 0.99 | 0.92             | 2.45  | 1.44 |
| *lf*  | 1.25     | 1.25        | 0.91        | 1.10 | 1.10 | 1.01             | 2.22  | 1.39 |
| *lg*  | 0.39     | 0.42        | 0.23        | 0.44 | 0.36 | 0.31             | 1.21  | 0.48 |

species which otherwise have very similar external morphologies (*Eberhard, 1985*; *Rowe & Arnqvist, 2011*). This contrast—generalised stabilising selection but fast macroevolutionary diversification—raises the intriguing question of how divergence in genital characters is initiated and amplified to reach the extraordinary variability observed in many groups (*Simmons et al., 2009*; *Hosken & Stockley, 2004*).

Most of the work on the evolution of genitalia has been conducted on reduced groups of species at limited temporal scales. There is very little quantitative data on the rate and mode of evolution of genital traits in diverse, old lineages, which could help to understand the origin of the large-scale variation in genital morphology (*Hosken & Stockley, 2004*; *Simmons, 2014*). In this work, we study the macroevolution of the size and complexity of male genitalia in a diverse insect lineage with a very uniform external morphology but extraordinary male genital variability, the aquatic beetle genus *Limnebius* (family Hydraenidae). *Limnebius* is divided in two sister lineages, one (subgenus *Bilimneus*) with uniformly small species (0.8–1.1 mm) with very simple genitalia, while the species of the other (subgenus *Limnebius* sensu stricto, "s.str." onwards) are much more variable in size (0.9–2.5 mm) and structure of the male genitalia (Fig. 1; Table 1; *Perkins, 1980*; *Jäch, 1993*; *Rudoy, Beutel & Ribera, 2016*). We obtained different measures of the complexity and size of the male genitalia and the body size of males and females of a comprehensive representation of species of both subgenera to (i) determine the phylogenetic signal and possible correlated evolution of the studied traits; (ii) compare their evolutionary rates; and (iii) estimate the distribution of the reconstructed phenotypic change through the phylogeny. Our expectation is that the comparison of the rate and mode of evolution of the size and complexity of the male genitalia between two lineages with strikingly contrasting macroevolutionary trends will contribute to understand the origin of their extraordinary diversity.

## MATERIALS AND METHODS

### Morphological measurements

*Limnebius* includes ca. 150 species with an almost cosmopolitan distribution, all of them aquatic, living in all types of continental waters with the only exception of saline habitats

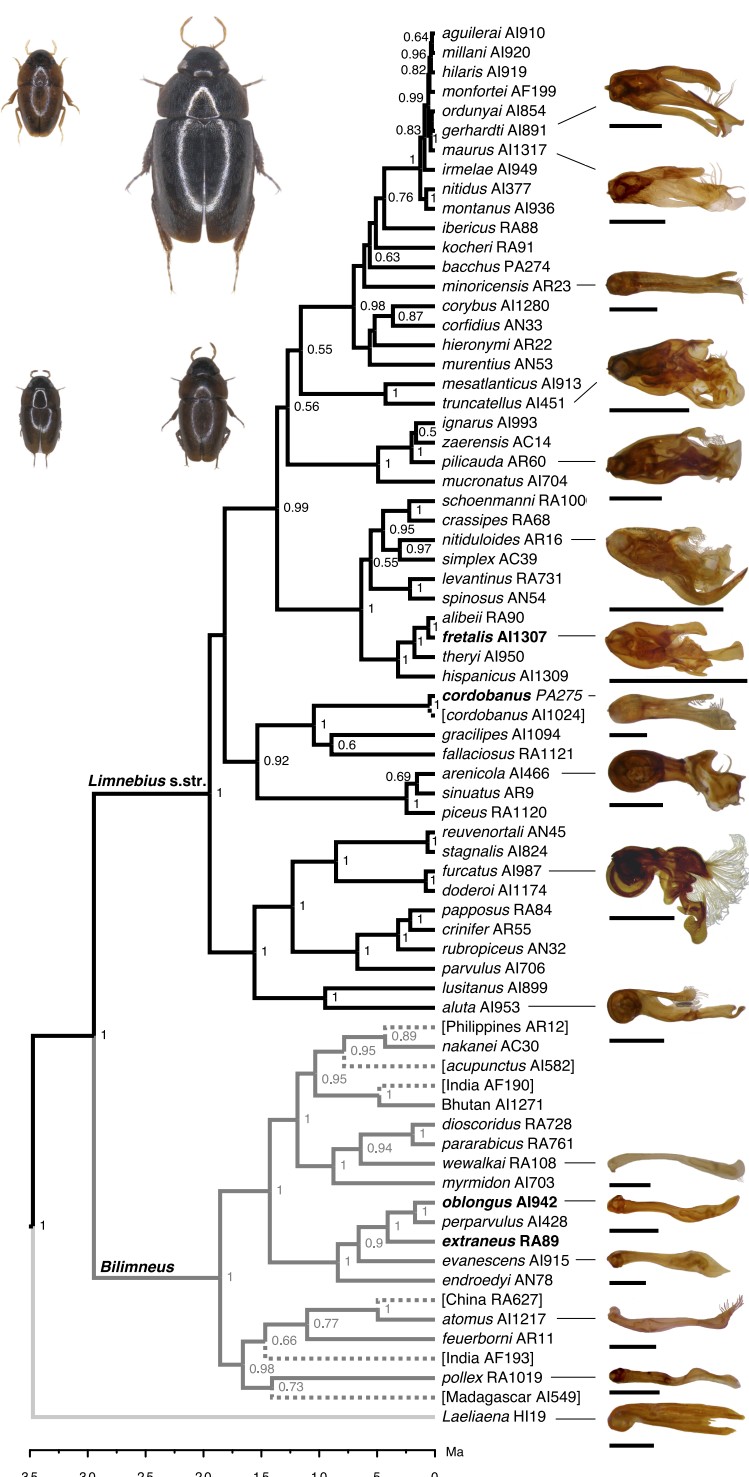

**Figure 1  Reconstructed phylogeny of the genus *Limnebius*, with estimated divergence times (Ma).** Dashed lines, terminal branches of species with no quantitative data for the aedeagus, deleted in the analyses (see text). Numbers in nodes, posterior probability as obtained in BEAST. Habitus photographs reflect maximum size variation within each subgenera; upper row: *L. cordobanus* and *L. fretalis*, lower row: *L. extraneus* and *L. oblongus* (marked in bold in the tree). Photos of the aedeagus in the right column standardised to the same size; scales refer to the size of the aedeagus of *L. fretalis* (the largest, 1.21 mm).

(*Perkins, 1980*; *Jäch, 1993*; *Hansen, 1998*; *Rudoy, Beutel & Ribera, 2016*). There is very few information on the ecology and behaviour of the species of *Limnebius*, including their sexual habits. In a recent work, *Limnebius* has been shown to be divided in two sister lineages with an estimated Oligocene origin, the subgenera *Bilimneus* and *Limnebius* s.str., with ca. 60 and 90 described species respectively (*Rudoy, Beutel & Ribera, 2016*).

We characterised the size and complexity of the male genitalia of all species included in the molecular phylogeny (see below and Table S1). The structure of the female genitalia of *Limnebius* is poorly known, mostly due to the lack of sclerotized parts (*Rudoy, Beutel & Ribera, 2016*), and was not studied here. We measured body length of adults (males, *lm* and females, *lf*) as the sum of the individual maximum lengths of pronotum and elytra, as the different position of the articulation between the two could alter the total length when measured together. Similarly, the head was not measured, as in many specimens it was partly concealed below the pronotum. Measures were obtained with stereoscope microscopes equipped with an ocular micrometer.

Male genitalia (aedeagi) were dissected and mounted on transparent labels with dimethyl hydantoin formaldehyde (DMHF). For size measurements we used as a single value the average of each measure in all studied specimens of the same species (Table S1). For the measures of complexity a single specimen was used, as species show in general a very constant shape of the aedeagus (*Jäch, 1993*; *Rudoy, Beutel & Ribera, 2016*), and there is no evidence for the widespread existence of cryptic species within the genus (unpublished data). Male genitalia were always photographed in the same standard positions, orientated according to the foramen in ventral and lateral views. We measured the maximum length of the male genitalia (*lg*) on digital images obtained in the same standard position, orientated in ventral view as determined by the foramen (*Rudoy, Beutel & Ribera, 2016*). We did not include setae or apical membranous structures, but included appendages when they were longer than the median lobe (as in e.g., some species of the *L. nitidus* group, *Rudoy, Beutel & Ribera, 2016*). Measurements were directly obtained from the digital images using ImageJ v.1.49 (US National Institutes of Health: http://imagej.nih.gov/ij/) (Fig. S1). We estimated experimental error by measuring the same specimen of three species on three different sessions, using two sets of images.

We used two different measures to characterise the complexity of the aedeagus:

(i) Perimeter (*per*) of the aedeagus in ventral view, including the median lobe and the main appendages. We obtained an outline of the genitalia from digital images using ImageJ, with the "free hand outline" option. The total perimeter was the sum of the values of the different parts of the genitalia (median lobe and left parameter, plus main appendages if present, see *Rudoy, Beutel & Ribera, 2016*). We standardised the values by dividing the perimeter by the length of the aedeagus, to obtain a measure of complexity by unit of length (Fig. S1 and Table S1).

(ii) Fractal dimension (*fd*). We estimated the fractal dimension of the outline of the aedeagus in ventral view on images of standard size (2100x2100 pixels, 2000 pixels from base to apex of the aedeagus) with the software Fractal Dimension Estimator (http://www.fractal-lab.org/index.html). This software estimates the Minkowski fractal dimension of bidimensional images using the box-counting method (*Falconer, 1990*). The

software converts the image to binary data, selects the scaling window of the box, and counts how many boxes are necessary to cover the image. The absolute value of the slope of a log–log graph of the scale with the number of boxes is the fractal dimension of the image (Fig. S1, Table S1).

## Phylogenetic analyses

We obtained a phylogeny of *Limnebius* using molecular data obtained from *Rudoy, Beutel & Ribera (2016)*, which included 71 of the ca. 150 known species of the genus. The taxon sampling was denser for the Palaearctic lineages in subgenus *Limnebius* s.str., including the full range of body sizes and aedeagus structural variation (*Rudoy, Beutel & Ribera, 2016*). Groups with the lowest number of sampled species show in general a more homogeneous and simpler aedeagus (*L. mundus* and *L. piceus* groups and subgenus *Bilimneus*). We used as outgroup and to root the tree the genus *Laeliaena*, considered to be sister to *Limnebius* based on multiple morphological synapomorphies (*Hansen, 1991*; *Jäch, 1995*; *Perkins, 1997*; *Beutel, Anton & Jäch, 2003*).

We reconstructed a phylogeny of the genus with Bayesian analyses in BEAST 1.8 (*Drummond et al., 2012*) using a combined data matrix with three partitions, (i) mitochondrial protein coding genes (two *cox1* fragments plus *nad1*); (ii) mitochondrial ribosomal genes (*rrnL* plus *trnL*) and (iii) nuclear ribosomal genes (*SSU* plus *LSU*) (Table S1). We used a Yule speciation process as the tree prior and an uncorrelated relaxed clock. Analyses were run for 100 MY generations, ensuring that the number of generations after convergence were sufficient as assessed with Tracer v1.6 (*Drummond et al., 2012*) and after removal of a conservative 10% burn-in fraction.

To calibrate the phylogenetic tree we used the rates estimated in *Cieslak, Fresneda & Ribera (2014)* for a related group (familiy Leiodidae, within the same superfamily Staphylinoidea, *Beutel & Leschen, 2005*) and the same gene combination based on the tectonic separation of the Sardinian plate 33 Ma. We set as prior average mean rate a normal distribution with average 0.015 substitutions/site/Myr for the mitochondrial protein genes, 0.006 for the mitochondrial ribosomal genes, and 0.004 for the nuclear ribosomal genes, all with a standard deviation of 0.001. It must be noted that for our objectives only relative rates are needed. An absolute calibration would only be necessary to obtain absolute estimates of character change, which is not our main objective and does not affect our conclusions.

We reconstructed the ancestral values of the morphological variables using the values of the terminals (extant species) in BEAST 1.8. The reconstruction of ancestral characters should ideally include a combination of extant and fossil data (e.g., *Webster & Purvis, 2002*; *Slater, Harmon & Alfaro, 2012*; *Bokma et al., 2015*). However, as there are no known fossils of species of *Limnebius* we can only use extant species. The use of extant species without fossil data has resulted in valuable contributions to the study of character evolution (e.g., *Cooper & Purvis, 2010*; *Baker et al., 2015*), and in some cases it has been reported that the introduction of fossil data did not alter substantially the conclusions obtained with extant species only (e.g., *Puttick & Thomas, 2015*).

We implemented a Brownian movement model of evolution (BM), a null model of homogeneous evolution in which variation accumulates proportionally with time, with incremental changes drawn from a random distribution with zero mean and finite constant variance (*Hunt & Rabosky, 2014*; *Adams, 2014*). The reconstruction of ancestral values using a BM model of evolution can be biased toward average or intermediate values (*Pagel, 1999*; *Finarelli & Goswami, 2013*), which together with the lack of fossil data can result in an underestimation of the rates of evolution of some characters. Due to these limitations our reconstruction needs to be understood as the simplest null model explaining the evolutionary change in the studied characters. It must be noted though that some of our conclusions do not depend on these reconstructions, and for others we compare two groups (*Bilimneus* and *Limnebius*) likely affected by the same biases. In any case, all analyses were repeated using only the data of the terminal branches, with a decreased power due to their lower number but less prone to reconstruction biases. Terminal branches are truncated with respect to internal branches, as they do not reflect a full interval between speciation events but the time elapsed since the last cladogenetic split. This does not introduce any bias in our measures when change is anagenetic (i.e., proportional to time), but will overestimate evolutionary rates if change occurs mostly at speciation (i.e., according to punctuated models of evolution; *Gingerich, 2009*; *Hunt & Rabosky, 2014*).

## Statistical analyses

We studied the evolution of the morphological characters trough the full evolutionary path of species (i.e., from root to tips) and in the individual branches, using phylogenetic ancestor-descendant comparisons (PAD; *Baker et al., 2015*). We first deleted the terminals with species with unknown males, as well as the duplicated specimen of *L. cordobanus* and the outgroup (*Laeliaena*) (Table S1). We then estimated the phylogenetic signal of the morphological variables in the whole tree using the $K$ metric proposed by *Blomberg, Garland & Ives (2003)*, which tests whether the topology and branch lengths of a given tree better fits a set of tip data compared with the fit obtained when the data have been randomly permuted. The higher the $K$ statistic, the more phylogenetic signal in a trait. $K$ values of 1 correspond to a BM model, which implies some degree of phylogenetic signal. $K$ values closer to zero correspond to a random or convergent pattern of evolution, while $K$ values greater than 1 indicate strong phylogenetic signal. We used the R package 'Picante' (*Kembel et al., 2010*) to compute $K$ and the significance test. To test the possible decrease of power or the randomization test due to the low number of species in *Bilimneus* (15, vs. 50 in *Limnebius* s.str.) we randomly pruned the tree of *Limnebius* s.str. to 15 species in 1,000 replicas and computed the $K$ values for all of them. We also measured the correlation between the variables across the whole tree with a regression of phylogenetic independent contrasts with the PDAP package in MESQUITE (*Midford, Garland & Maddison, 2011*).

For the analyses of the individual branches of the tree (PADs) we compiled the reconstructed values in BEAST of all variables for the initial and final node of each individual branch, together with their length. We measured three values for each of the individual branches (including terminals): (i) amount of phenotypic change, equal to the arithmetic difference between the final and initial values of the branch; (ii) absolute amount

of phenotypic change, equal to the absolute value of the amount of phenotypic change; (iii) phenotypic change measured in darwins (*Haldane, 1949*), computed as the absolute value of the natural logarithm of the ratio between the final and initial values divided by the length of the branch in million years (Myr) (Table S1). The use of the natural logarithm standardises the change so it is proportional and directly comparable among species with different sizes (*Haldane, 1949*; *Gingerich, 2009*). Darwins are known to be strongly dependent on the length over which are measured (*Gingerich, 2009*; *Hunt, 2012*; *Hunt & Rabosky, 2014*), so to obtain the correlation with time of the different variables we used instead the absolute amount of phenotypic change, which showed a more linear relationship with branch length for most of the variables (see Results). As individual branches are in principle independent from each other we analysed these variables with standard statistical procedures (see e.g., *Baker et al., 2015*). We used a Bonferroni correction for multiple tests and considered as significant a $p < 0.01$ level (corresponding to five morphological variables), although we also report some results at the standard $p < 0.05$ level in some of the analyses. In the comparisons between the two subgenera of *Limnebius* (*Bilimneus* and *Limnebius* s.str.) we did not include the respective stem branches.

## RESULTS

### Phylogenetic signal and correlation between complexity and size

The reconstructed ancestral values of *Bilimneus* were in the upper range of the values of the extant species, especially for measures of complexity (*fd* and *per*). The reconstructed ancestral values of the whole genus were, on the contrary, larger than any extant species of *Bilimneus*, with the only exception of the length of the genitalia (*lg*) (Table 1). For the subgenus *Limnebius* s.str. all reconstructed values, both of the genus and the subgenus, were within the range of the extant species, and close to their centre of distribution (Table 1; Tables S1 and S2).

The phylogenetic signal ($K$) was very significant for all variables both in the genus *Limnebius* as a whole and the subgenus *Limnebius* s.str., with values close to 1 (i.e., a BM model) for the measures of complexity (*per* and *fd*), clearly over 1 (i.e., with a stronger phylogenetic signal) for the size of males and females (*lf* and *lm*), but with values lower than 1 (i.e., closer to a global random distribution, although still significant) for the length of the genitalia (*lg*) (Table 2).In contrast, none of the variables had a significant $K$ in subgenus *Bilimneus* (Table 2). More than 99% of the 1,000 pruned replicas of *Limnebius* s.str. maintained significant values of $K$ for all variables at a $p < 0.05$ level, and more than 90% at $p < 0.01$, with the only exception of *lg*, which become not significant (Table 2).

All pairwise correlations between the morphological variables were significant at a $p < 0.01$ level as measured with PDAP both in *Limnebius* as a whole and in subgenus *Limnebius* s.str., with the exception of the correlation between *lg* and the measures of complexity (*per* and *fd*) (Table 3). For *Bilimneus*, with a lower number of contrasts (14 vs 49 in *Limnebius* s.str.), results were similar except that the length of the female (*lf*) was not significantly correlated with any variable, and the length of the aedeagus (*lg*) was significantly correlated with its fractal dimension (Table 3).

Table 2   Values of the *K* statistic for all measured variables in the genus *Limnebius* and its two subgenera, including the randomized pruned 1,000 replicas of the tree of *Limnebius* s.str. with only 15 species (**see text**). See Table 1 for the codes of the measured variables.

| | *Limnebius* | *Bilimneus* | *Limnebius* s.str. | 15 spp × 1000 avg | var |
|---|---|---|---|---|---|
| *per* | 1.24** | 0.24 | 0.97** | 1.04* | 0.08 |
| *fd* | 1.55** | 0.50 | 1.19** | 1.04* | 0.06 |
| *lf* | 1.65** | 0.67 | 1.51** | 1.23** | 0.06 |
| *lm* | 1.35** | 0.24 | 1.34** | 1.25** | 0.06 |
| *lg* | 0.63** | 0.53 | 0.71** | 0.74 | 0.03 |

**Notes.**
*Significant values at a $p < 0.05$.
**Significant values at a $p < 0.01$.

Table 3   **Correlation between the measured variables, as estimated with phylogenetic independent contrasts.** See Table 1 for the codes of the measured variables.

| | | *per* | *fd* | *lf* | *lm* |
|---|---|---|---|---|---|
| *Limnebius* | *fd* | 0.75** | | | |
| $n = 65$ | *lf* | 0.44** | 0.48** | | |
| | *lm* | 0.53** | 0.50** | 0.87** | |
| | *lg* | 0.11 | 0.31 | 0.53** | 0.53** |
| *Bilimneus* | *fd* | 0.66** | | | |
| $n = 14$ | *lf* | −0.04 | 0.02 | | |
| | *lm* | 0.75** | 0.73** | 0.38 | |
| | *lg* | 0.41 | 0.70** | 0.08 | 0.71** |
| *Limnebius* s.str. | *fd* | 0.76** | | | |
| $n = 49$ | *lf* | 0.43** | 0.47** | | |
| | *lm* | 0.52** | 0.48** | 0.89** | |
| | *lg* | 0.10 | 0.29 | 0.54** | 0.52** |

**Notes.**
**Significant values at a $p < 0.01$.

## Rates of evolution

There were no significant differences between terminal and all branches of the tree for any of the measured variables (including branch length) at a $p < 0.01$ level, both in average value (2 tailed t-Student with unequal variances when differences were significant) or variance (Fisher's F-test) (Table 4). At a $p < 0.05$ only for the measure of darwins the variances of *per* and *lf* were larger in the terminals than in all branches, and the variance of *lg* smaller (Table 4). Some of the largest increases or decreases of absolute change were in terminal branches (e.g., *L. nitiduloides* and *L. cordobanus* respectively, branches 94 and 72 in Fig. S2).

The variable with the highest values of phenotypic change and the highest variance was *lg*, both measured in absolute phenotypic change or in darwins, and both in *Bilimneus* and *Limnebius* s.str. (Table 4). The lowest values of change were in *fd*, also in both subgenera. Males were in general more labile than females, with larger relative (but not absolute) differences in *Bilimneus* than in *Limnebius* s.str. (Table 4).

**Table 4 Average value and variance of the average of the measures of evolutionary change in the individual branches of the phylogeny (PAD analyses; see Methods).** Measures in darwins ×10(−6). See Table 1 for the codes of the measured variables, and Fig. S3 for a graphic estimation of the density functions of the different comparisons.

| | Terminals | | All branches | | Bilimneus | | Limnebius s.str. | |
| --- | --- | --- | --- | --- | --- | --- | --- | --- |
| | avg | var | avg | var | avg | var | avg | var |
| b.l. | 4.61 | 28.75 | 3.47 | 11.16 | 5.50** | 4.29** | 2.89** | 2.78** |
| phenotypic change | | | | | | | | |
| per | −0.071 | 0.358 | 0.005 | 0.360 | −0.068 | 0.015** | 0.026 | 0.458** |
| fd | −0.002 | 0.000 | 0.000 | 0.000 | −0.004* | 0.000** | 0.001* | 0.000** |
| lf | −0.007 | 0.004 | 0.000 | 0.006 | −0.009 | 0.001** | 0.003 | 0.008** |
| lm | −0.014 | 0.012 | 0.000 | 0.013 | −0.017 | 0.004** | 0.005 | 0.016** |
| lg | −0.009 | 0.004 | 0.000 | 0.003 | −0.003 | 0.002* | 0.002 | 0.004* |
| absolute phenotypic change | | | | | | | | |
| per | 0.423 | 0.182 | 0.373 | 0.220 | 0.102** | 0.009** | 0.450** | 0.253** |
| fd | 0.014 | 0.000 | 0.013 | 0.000 | 0.006** | 0.000** | 0.015** | 0.000** |
| lf | 0.045 | 0.002 | 0.048 | 0.004 | 0.022** | 0.000** | 0.055** | 0.004** |
| lm | 0.082 | 0.005 | 0.075 | 0.008 | 0.048** | 0.001** | 0.083** | 0.009** |
| lg | 0.047 | 0.002 | 0.040 | 0.002 | 0.033 | 0.001** | 0.043 | 0.002** |
| darwins | | | | | | | | |
| per | 0.048 | 0.005* | 0.039 | 0.003* | 0.011** | 0.000** | 0.048** | 0.003** |
| fd | 0.005 | 0.000 | 0.005 | 0.000 | 0.002** | 0.000** | 0.006** | 0.000** |
| lf | 0.019 | 0.001* | 0.017 | 0.001* | 0.006** | 0.000** | 0.021** | 0.001** |
| lm | 0.030 | 0.001 | 0.023 | 0.001 | 0.014* | 0.000** | 0.026* | 0.001** |
| lg | 0.047 | 0.002 | 0.043 | 0.003 | 0.022 | 0.001** | 0.050 | 0.004** |

**Notes.**
*Significant values at a $p < 0.05$.
**Significant values at a $p < 0.01$.

All comparisons of absolute change and darwins between *Bilimneus* and *Limnebius* s.str. were significantly different (always lower in *Bilimneus*) both for the variance and the average, with the only exception of the average *lg*, never significantly different between them regardless of how it was measured (Table 4; Fig. S3). When the sign of the change was considered variances were still significantly different, but only the average *fd* showed differences (positive in *Limnebius* s.str. and negative in *Bilimneus*), although only at a $p < 0.05$ level (Table 4; Fig. S3).

Branches of *Limnebius* s.str. were on average significantly shorter and with a lower variance than that of *Bilimneus* (Table 4; Fig. S3). In the whole *Limnebius* and the subgenus *Limnebius* s.str. all the measures of absolute phenotypic change were significantly correlated with the length of the branch, with *lg* having the lowest correlation (Fig. 2A; Table 5). On the contrary, for *Bilimneus* only *lg* and *per* had significant correlations. When only the terminal branches were used the correlations between the measures of absolute phenotypic change and branch lengths were still significant at a $p < 0.01$ level for *Limnebius* s.str. with the only exception of *lg*, but none of them was significant for *Bilimneus* (Fig. 2B and Table 5).

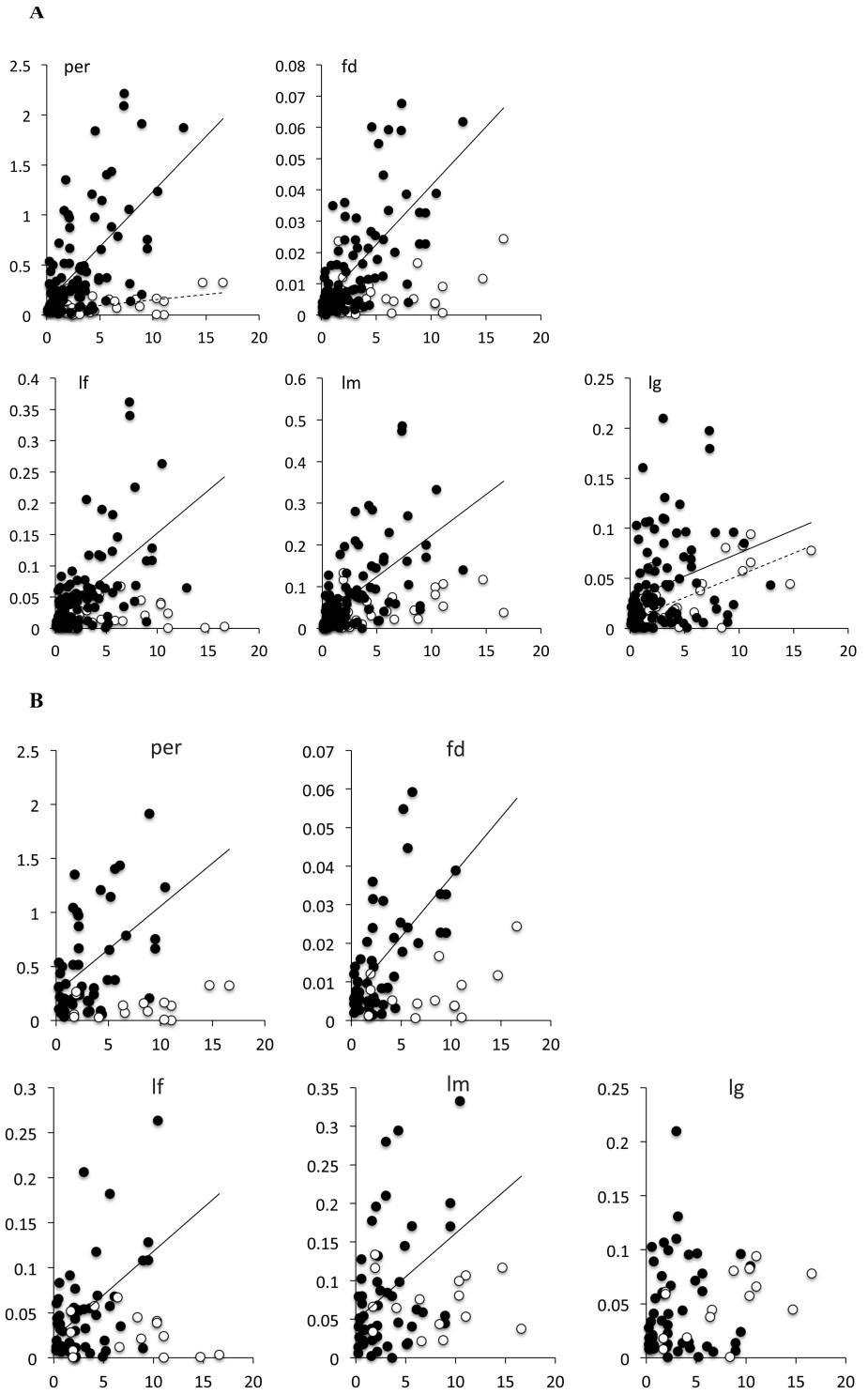

**Figure 2** **Bivariate plot of the branch length with the measure of phenotypic change for (A) all branches in the tree, and (B) terminal branches only.** Solid circles, species of *Limnebius* s.str.; open circles, species of *Bilimneus*. See Table 5 for the parameters of the linear regressions.

**Table 5  Parameters of the linear regression between branch length and the measures of absolute change for all and for terminal branches, and for the whole genus *Limnebius*, *Bilimneus* and *Limnebius* s.str.**  See Table 1 for the codes of the measured variables, and Fig. 2.

| All branches | R2 | p | Intercept | 95% intercept Min | 95% intercept Max | Slope | 95% Slope Min | 95% Slope Max |
|---|---|---|---|---|---|---|---|---|
| *Limnebius* | | | | | | | | |
| per | 0.121** | <0.0001** | 0.203 | 0.091 | 0.316 | 0.049 | 0.026 | 0.072 |
| fd | 0.184** | <0.0001** | 0.006 | 0.003 | 0.010 | 0.002 | 0.001 | 0.003 |
| lf | 0.105** | 0.000** | 0.027 | 0.012 | 0.042 | 0.006 | 0.003 | 0.009 |
| lm | 0.158** | <0.0001** | 0.039 | 0.019 | 0.059 | 0.010 | 0.006 | 0.015 |
| lg | 0.083** | 0.001** | 0.028 | 0.017 | 0.038 | 0.004 | 0.002 | 0.006 |
| *Bilimneus* | | | | | | | | |
| per | 0.237** | 0.009** | 0.041 | −0.014** | 0.096** | 0.011 | 0.003 | 0.019 |
| fd | 0.133 | 0.057 | 0.003 | 0.000 | 0.007 | 0.001 | 0.000 | 0.001 |
| lf | 0.001 | 0.910 | 0.023 | 0.010 | 0.035 | 0.000 | −0.002** | 0.002** |
| lm | 0.138 | 0.052 | 0.030 | 0.007 | 0.053 | 0.003 | 0.000** | 0.007** |
| lg | 0.443** | 0.000** | 0.009 | −0.005** | 0.023** | 0.004 | 0.002 | 0.006 |
| *Limnebius* sstr | | | | | | | | |
| per | 0.370** | <0.0001** | 0.132 | 0.015 | 0.248 | 0.110 | 0.081 | 0.139 |
| fd | 0.439** | <0.0001** | 0.004 | 0.000 | 0.007 | 0.004 | 0.003 | 0.005 |
| lf | 0.320** | <0.0001** | 0.016 | 0.000 | 0.032 | 0.014 | 0.010 | 0.018 |
| lm | 0.331** | <0.0001** | 0.026 | 0.003 | 0.048 | 0.020 | 0.014 | 0.025 |
| lg | 0.078** | 0.005** | 0.029 | 0.016 | 0.042 | 0.005 | 0.001 | 0.008 |
| **Terminal branches** | | | | | | | | |
| *Limnebius* | | | | | | | | |
| per | 0.015 | 0.326 | 0.362 | 0.205 | 0.519 | 0.014 | −0.014** | 0.042** |
| fd | 0.083* | 0.020* | 0.010 | 0.005 | 0.014 | 0.001 | 0.000 | 0.002 |
| lf | 0.024 | 0.216 | 0.037 | 0.019 | 0.055 | 0.002 | −0.001** | 0.005** |
| lm | 0.031 | 0.163 | 0.039 | 0.025 | 0.054 | 0.002 | −0.001** | 0.004** |
| lg | 0.038 | 0.118 | 0.065 | 0.040 | 0.091 | 0.004 | −0.001** | 0.008** |
| *Bilimneus* | | | | | | | | |
| per | 0.109 | 0.229 | 0.078 | −0.040** | 0.195** | 0.008 | −0.005** | 0.021** |
| fd | 0.236 | 0.067 | 0.002 | −0.005** | 0.009** | 0.001 | 0.000** | 0.001** |
| lf | 0.092 | 0.273 | 0.038 | 0.013 | 0.062 | −0.001 | −0.004** | 0.001** |
| lm | 0.006 | 0.786 | 0.076 | 0.035 | 0.117 | −0.001 | −0.005** | 0.004** |
| lg | 0.256 | 0.054 | 0.027 | −0.001** | 0.055** | 0.003 | 0.000** | 0.006** |
| *Limnebius* sstr | | | | | | | | |
| per | 0.241** | 0.000** | 0.258 | 0.088 | 0.428 | 0.080 | 0.039 | 0.121 |
| fd | 0.379** | <0.0001** | 0.006 | 0.002 | 0.011 | 0.003 | 0.002 | 0.004 |
| lf | 0.248** | 0.000** | 0.021 | 0.001 | 0.042 | 0.010 | 0.005 | 0.015 |
| lm | 0.156** | 0.004** | 0.049 | 0.017 | 0.080 | 0.011 | 0.004 | 0.019 |
| lg | 0.008 | 0.535 | 0.041 | 0.023 | 0.060 | 0.001 | −0.003** | 0.006** |

**Notes.**
*Significant values at a $p < 0.05$.
**Significant values at a $p < 0.01$.

Table 6 Kolmogorov–Smirnov test of fit to a normal distribution of the phenotypic change and darwins (considering the sign of the change) for all and terminal branches, and for *Bilimneus* and *Limnebius* s.str.

| | var | *Limnebius* | Terminal branches | *Bilimneus* | *Limnebius* sstr |
|---|---|---|---|---|---|
| Phenotpypic change | *per* | 0.006** | 0.20 | 0.70 | 0.06 |
| | *fd* | 0.002** | 0.19 | 0.18 | 0.01* |
| | *lf* | 0.02* | 0.45 | 0.79 | 0.11 |
| | *lm* | 0.06 | 0.69 | 0.75 | 0.06 |
| | *lg* | 0.05 | 0.69 | 0.75 | 0.05* |
| Darwins | *per* | 0.004** | 0.03* | 0.16 | 0.007** |
| | *fd* | 0.01* | 0.06 | 0.04* | 0.06 |
| | *lf* | 0.001** | 0.01* | 0.31 | 0.03* |
| | *lm* | 0.01* | 0.09 | 0.17 | 0.09 |
| | *lg* | 0.01* | 0.14 | 0.68 | 0.13 |

Notes.
*Significant values at a $p < 0.05$.
**Significant values at a $p < 0.01$.

## Distribution of the reconstructed phenotypic change

The amount of change in the individual branches for all variables was in general not significantly different from a normal distribution, as measured with a non-parametric Kolmogorov–Smirnov test. Only the variables of complexity when all the branches in the whole genus were considered could be said not to follow a normal distribution at a $p < 0.01$ level (Table 6). When measured in darwins (and taking into account the sign of the branch), the amount of change of the individual variables was in general less normally distributed than when the change was measured with independence of the time, although at a $p < 0.01$ level only *per* and *lf* in the whole genus, and *per* in *Limnebius* s.str., were significantly non-normal (Table 6).

Despite the general normal distribution of the amount of change, in *Bilimneus* there was an excess of negative branches (i.e., a change to lower values) for all variables except for *lg*. On the contrary, in *Limnebius* s.str. there were significantly more positive than negative changes for the variables of complexity (Table 7). In both cases differences were mostly significant only at a $p < 0.05$ level, and in *Limnebius* s.str. only when all branches of the tree were considered. The average amount of phenotypic change per branch was negative for all measures in *Bilimneus* and positive in *Limnebius* s.str., but in all cases the 95% confidence intervals included the zero (data not shown).

For all measurements the general tendency was for a branch to have the same type of change than that of the preceding. In *Bilimneus* this meant that the most frequent combination was two branches with negative change, while in *Limnebius* s.str. two with positive change, although *lg* had the lowest differences between the frequency of two consecutive positive or negative branches in both subgenera (Table 7). The terminal branches showed the same tendency except for the variables measuring length in *Limnebius* s.str., for which the most frequent combination was two negative branches (Table 7).

Table 7 **Number of branches with positive or negative change (A), and sign of a branch and the preceding one (B), for all and terminal branches, and the genus *Limnebius* and the two subgenera (*Bilimneus* and *Limnebius* s.str.).** all, all branches; T, terminal branches; p.b., sign of the previous branch.

| (A) Sign of the branch | | per + | per − | fd + | fd − | lf + | lf − | lm + | lm − | lg + | lg − |
|---|---|---|---|---|---|---|---|---|---|---|---|
| *Limnebius* | all | 64 | 64 | 66 | 62 | 62 | 66 | 58* | 70* | 61 | 67 |
| | T | 28 | 37 | 30 | 35 | 32 | 33 | 27* | 38* | 29 | 36 |
| *Bilimneus* | all | 8* | 18* | 7** | 19** | 9* | 17* | 8* | 18* | 13 | 13 |
| | T | 5 | 10 | 4* | 11* | 7* | 8* | 4* | 11* | 8 | 7 |
| *Limnebius* sstr | all | 54* | 42* | 56* | 40* | 51 | 45 | 48 | 48 | 45 | 51 |
| | T | 23 | 27 | 26 | 24 | 25 | 25 | 23 | 27 | 21 | 29 |

| (B) Sign of the branch vs. preceding | | p.b. | per + | per − | fd + | fd − | lf + | lf − | lm + | lm − | lg + | lg − |
|---|---|---|---|---|---|---|---|---|---|---|---|---|
| *Limnebius* | all | + | 45** | 19** | 46** | 20** | 41** | 21** | 39** | 19** | 41** | 20** |
| | | − | 29** | 35** | 26** | 36** | 21** | 45** | 23** | 47** | 25** | 42** |
| | T | + | 20* | 8* | 20* | 10* | 19** | 13** | 18** | 9** | 15 | 14 |
| | | − | 17* | 20* | 17* | 18* | 10** | 23** | 14** | 24** | 15 | 21 |
| *Bilimneus* | all | + | 3* | 5* | 2* | 5* | 3** | 6** | 4* | 4* | 7* | 6* |
| | | − | 3* | 15* | 4* | 15* | 1** | 16** | 4* | 14* | 3* | 10* |
| | T | + | 3* | 2* | 1 | 3 | 3 | 4 | 3 | 1 | 4 | 4 |
| | | − | 1* | 9* | 3 | 8 | 1 | 7 | 4 | 7 | 2 | 5 |
| *Limnebius* sstr | all | + | 40* | 14* | 42** | 14** | 36** | 15** | 34** | 14** | 31** | 14** |
| | | − | 24* | 18* | 22** | 18** | 18** | 27** | 18** | 30** | 21** | 30** |
| | T | + | 17 | 6 | 19 | 7 | 16* | 9* | 15* | 8* | 11* | 10* |
| | | − | 16 | 11 | 14 | 10 | 9* | 16* | 10* | 17* | 13* | 16* |

**Notes.**
*Significant values at a $p < 0.05$.
**Significant values at a $p < 0.01$.

## DISCUSSION

We found clear differences in the macroevolution of the size and complexity of the aedeagus between the two subgenera of *Limnebius*. In the subgenus with the highest diversity of body size and aedeagus structure (*Limnebius* s.str.), both measures of complexity and the body size of males and females had a strong phylogenetic signal. This signal was much lower for the size of the genitalia, which evolved independently of complexity. Both measures of complexity had values of the $K$ statistic closer to 1, suggesting a BM model of evolution (*Blomberg, Garland & Ives, 2003*). This was supported by the strong correlation of both variables with the length of the branch (a surrogate of time), as expected of characters evolving under a random walk (*Gingerich, 2009*; *Hunt, 2012*; *Hunt & Rabosky, 2014*).

In the subgenus with the more homogeneous and simple genitalia, *Bilimneus*, the tendency was for all measured variables to evolve more randomly, despite the significance of some correlations. The correlation of the aedeagus size with its fractal dimension may be due allometry resulting from its simplicity and the higher dimensionality of the variable *fd*. In *Limnebius* s.str this allometric effect may be hidden by the larger variation in complexity.
In *Bilimneus*, length and perimeter of the genitalia were the only variables significantly correlated with time, but in both cases the effect size was very low, and very similar in both subgenera for *lg* (Fig. 2). The amount of phenotypic change in *Bilimneus* for all variables was also smaller and with a much smaller variance than in *Limnebius* s.str., except for *lg*. Despite the large difference in size between some species between the two subgenera, there were no significant differences in the average amount of change of the length of the genitalia in the individual branches, both with the measure of absolute phenotypic change or in darwins.

There was some evidence for directional evolution in the complexity of the genitalia in both subgenera, as revealed by the excess of branches with a positive change in *Limnebius* s.str. and negative in *Bilimneus*, both individually and in combination to other branches of the same sign. But this evidence for directional evolution was weak, and the fact that in all cases the 95% interval of the average of the measure of change included zero suggests that the dominant factor was a random walk (*Hunt, 2012*). The directionality could also be partly due to the effect of the hard boundary imposed by the intermediate condition of the reconstructed ancestor or the artefacts introduced by the reconstruction method (*Pagel, 1999*; *Hunt & Rabosky, 2014*). These biases for negative or positive change were, however, not present for the length of the genitalia, which was more normally distributed and with a more balanced distribution of positive and negative changes. Directional phenotypic change is rare in the fossil record, accounting for a small fraction of the studied systems, likely due to the short time scale in which directional selection needs to act to reach adaptive peaks (*Hunt, 2007*; *Hunt & Rabosky, 2014*). The evolution of most lineages seem to be driven by a complex interaction of random walk, stasis and directional evolution, suggesting bouts of different types of evolutionary constraints in an idiosyncratic sequence (*Hunt, 2012*; *Hunt, Hopkins & Lidgard, 2015*). Experimental results on the effect of sexual selection agree with this idiosyncratic view of the macroevolution of the genitalia. Selection can act differently on different genital traits, but there is no evidence of the existence of directional trends acting through long evolutionary scales which may accelerate macroevolutionary divergence (e.g., *House & Simmons, 2003*; *Simmons et al., 2009*; *Macagno et al., 2011*; *Rowe & Arnqvist, 2011*; *Simmons & García-González, 2011*; *Hotzy et al., 2012*; *Sakurai, Himuro & Kasuya, 2012*; *Okuzaki & Sota, 2014*; *Richmond, 2014*; *Dougherty et al., 2015*).

Contrary to what happened with the evolution of complexity, the weaker phylogenetic signal in the length of the male genitalia, weaker correlation with time and more normal distribution of change would be compatible with an evolution dominated either by random factors or by stasis (*Hunt, 2012*; *Hunt & Rabosky, 2014*). This would seem paradoxical, as genital length is the variable with the highest evolutionary rates in both subgenera, but it should be considered that stasis is defined by the bias of the phenotypic variation of a trait, not by its amplitude—that is, for the tendency of the trait to return to an overall optimal value (*Hunt, 2012*), but not for the amplitude of variation around this optimum. It must also be considered that in our PAD analyses we pooled branches from multiple independent lineages, which means that even when the length of the genitalia may show a macroevolutionary pattern compatible with stasis and stabilizing selection, it may still be subjected to directional selection in different lineages.

The main difference in the evolution of the male genitalia between the two subgenera seems thus to be the strong contrast in the variance of the amount of phenotypic change. Whether this difference was due to a constraint in the variation in *Bilimneus* or a relaxation in *Limnebius* s.str. is at present unknown. The low variability of the size and complexity of the genitalia within the sister genus of *Limnebius*, *Laeliaena* (*Jäch, 1995*; *Rudoy, Beutel & Ribera, 2016*) suggest the possibility that the higher variability of *Limnebius* s.str. is the derived condition, but the low number of species of *Laeliaena* (three) precludes any firm conclusion. A potential key factor could be the different relative size of males and females in the two subgenera of *Limnebius*: while in *Limnebius* s.str. males are usually larger (sometimes much larger) than females, in *Bilimneus* females are larger than males (Table S1), suggesting a different role of sexual selection in the two subgenera.

There are two main potential biases affecting our conclusions, the lack of fossil data and the use of a BM model in the reconstruction of ancestral values. As already noted, the lack of fossil data in combination with a BM reconstruction model imply that all reconstructed values (including the ancestors) should be within the range of the extant species, which may not be accurate (*Pagel, 1999*). This may result in an underestimation of the variance and of the number of shifts in the direction of change, i.e., a stronger tendency to linearity—but also a stronger tendency to find that variables evolve under a BM model, which may led to some degree of circularity. The analyses of the phylogenetic signal ($K$) and the overall correlation between variables were, however, not affected by these biases, but still clearly reflected differences between subgenera and between the evolution of complexity and size of the aedeagus. Similarly, the results of the PAD analyses using the terminal branches were in general in agreement with that of all the branches of the tree, although with less statistical power due to their lower number.

## CONCLUSIONS

The extraordinary variation in body size and male genitalia in one of the lineages of the genus *Limnebius* seems to be the result not of a clear directional trend, but of its larger macroevolutionary variance. The complexity of the aedeagus in *Limnebius* s.str. largely evolved in good agreement with a Brownian motion model, although we found evidence of weak directional selection for respectively and increase or decrease of complexity in the two subgenera, likely acting in some lineages and at limited temporal scales. In contrast, change in the size of the genitalia, the variable with the highest rate of phenotypic evolution, was less dependent on the phylogeny and time, and without any evidence of directional selection.

## ACKNOWLEDGEMENTS

We are grateful to MA Jäch (NMW, Wien) and P Perkins (MCZ, Harvard) for allowing us to study the collections of their institutions and for supporting this work in many ways, and A Cieslak, A Cordero (University of Vigo) and two anonymous Referees for comments to the manuscript. AR thanks MA Jäch for his help during his stay in the NMW.

### Funding

This work was partly funded by a JAE PhD studentship (CSIC) to AR, the Spanish Ministerio de Economía y Competitividad (projects CGL2010-15755 and CGL2013-48950-C2-1-P) and a Salvador de Madariaga grant in the Phyletisches Museum in Jena (PRX14/00583) to IR The funders had no role in study design, data collection and analysis, decision to publish, or preparation of the manuscript.

### Grant Disclosures

The following grant information was disclosed by the authors:
JAE PhD studentship (CSIC).
Spanish Ministerio de Economía y Competitividad: projects CGL2010-15755 and CGL2013-48950-C2-1-P.
Phyletisches Museum in Jena: PRX14/00583.

### Competing Interests

The authors declare there are no competing interests.

### Author Contributions

- Andrey Rudoy and Ignacio Ribera conceived and designed the experiments, performed the experiments, analyzed the data, contributed reagents/materials/analysis tools, wrote the paper, prepared figures and/or tables, reviewed drafts of the paper.

### Data Availability

The raw data has been supplied as Supplemental Dataset files (Tables S1 and S2).

### Supplemental Information

Supplemental information for this article can be found online at http://dx.doi.org/10.7717/peerj.1882#supplemental-information.

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
