# Peer review of "The macroevolution of size and complexity in insect male genitalia"

_PeerJ, doi:10.7717/peerj.1882_

## Round 0.1 · original submission · Minor Revisions

Both reviewers had minor corrections, and I'm confident these minor points can be addressed quickly. If you can address these, we can move forward with the publication process.

Reviewer 1 ·

Basic reporting

The paper meets the standards.

Experimental design

This paper uses a comprehensive phylogeny of one beetle genus, to address the evolution of genital size and complexity. The goals are clearly stated and are relevant. Methods are well described, but I do have some suggestions:
(1) The size of the images of genitalia in Figure 1 is unavoidably small. Therefore, I think that including a few images at larger resolution, indicating how measurements were done, as supplementary files, would be needed, particularly for future work which try to use similar methods.
(2) Line 136. How much was the burn-in period?

Validity of the findings

Data are included as supplementary files

Additional comments

Some minor comments:
(3) Lines 62-65. I wonder how different are species in other characters rather than genitalia. Does the subgenus Bilimneus include species which can be labelled as cryptic? Are these species diferentiated by ecology or behaviour?
(4) Line 106. The web page for ImageJ has been already included in line 100.
(5) The scale of images in Figure 1 refers to the size of the aedeagus of the largest species, but it is unclear to me what this mean...
(6) Figure 2. There are open and closed dots, which are not explained
(7) A general question. Is there information on how female genitalia differs between species?

Reviewer 2 ·

Basic reporting

No comments

Experimental design

Clear and appropriate

Validity of the findings

Clearly validated

Additional comments

An interesting paper, extending our understanding of a significant evolutionary topic. My only suggestion is that the take-home messages could be better summarized in two places - the end of the abstract, and the conclusions section at the end. Both of these as currently written could be a little difficult to follow for a general reader (who will, I suspect, be interested in such a paper).

Specifically:

"...The origin of the larger variation in Limnebius s.str. seems to be the significantly larger average and variance of the reconstructed values of change in the individual branches of the tree for all measured variables, except for the average change in genital length, not significantly different between the two subgenera." should be rewritten. e.g. 'translate' "significantly larger average and variance of the reconstructed values of change in the individual branches of the tree for all measured variables" into something which is less 'statistical' and more 'biological'.

Similarly I think the conclusion should be revisited. Such changes make the overall message more accessible in my opinion.

---

## Round 0.2 · accepted · Accept

Thanks for your close attention to the reviewers comments. I think the revised manuscript reads well, and presents an intriguing story.